# Incidence and Risk Factors of White Matter Lesions in Moderate and Late Preterm Infants

**DOI:** 10.3390/diagnostics15070881

**Published:** 2025-04-01

**Authors:** Kentaro Ueda, Kennosuke Tsuda, Takaharu Yamada, Shin Kato, Sachiko Iwata, Shinji Saitoh, Osuke Iwata

**Affiliations:** 1Department of Paediatrics, Japanese Red Cross Aichi Medical Centre Nagoya Daini Hospital, Nagoya 466-8650, Japan; 2Centre for Human Development and Family Science, Department of Paediatrics and Neonatology, Nagoya City University Graduate School of Medical Sciences, Nagoya 464-0083, Japano.iwata@med.nagoya-cu.ac.jp (O.I.)

**Keywords:** preterm infants, white matter, brain lesions, magnetic resonance imaging

## Abstract

**Background**: Moderate and late preterm infants (32–36 weeks of gestation) are at significant risk of developmental impairments. Incidence of white matter lesions, which are associated with developmental impairments in very preterm infants, remains underreported in this population. This study aimed to assess the incidence and clinical risk factors associated with brain lesions, particularly white matter lesions, in moderate and late preterm infants using term-equivalent MRI. **Methods**: This prospective observational study included 195 preterm infants born at 32+0–36+6 weeks of gestation and admitted to a tertiary NICU between 2019 and 2020. MRI findings at term-equivalent age were evaluated. Clinical risk factors were analysed using logistic regression. **Results**: Among the 195 infants, 23.6% had brain lesions on MRI, with white matter lesions (73.9%), specifically punctate white matter lesions, being the most common form of lesions. Vaginal delivery (odds ratio (OR) = 3.102, 95% confidence interval (CI) = 1.250–7.696, *p* = 0.015), larger birth weight z-scores (OR = 1.702, 95% CI = 1.118–2.591, *p* = 0.013), and intubation (OR = 2.948, 95% CI = 1.269–6.850, *p* = 0.012) were significant risk factors for white matter lesions. **Conclusions**: White matter lesions, particularly punctate white matter lesions, are common in moderate and late preterm infants. These lesions are associated with perinatal factors suggestive of delayed transition and inflammation. Future research should focus on detailed clinical care measures and neurodevelopmental assessments to identify modifiable risk factors for brain injury.

## 1. Introduction

Approximately 10–15% of very preterm infants (<30 weeks of gestation) develop cerebral palsy, 40% develop mild motor deficits, and 30–60% experience cognitive deficits [1]. Increasing evidence suggests that moderate and late preterm infants born at 32–36 weeks are at higher risk of developmental sequelae, including cerebral palsy (relative risk: 3.52–14.1), global developmental delay (relative risk: 1.61–2.89), and attention-deficit or hyperactivity disorder (relative risk: 1.25–1.62) [2].

Identifying brain lesions associated with developmental impairments and upstream risk factors is crucial for preventing adverse outcomes following preterm birth [3]. Intraventricular haemorrhage (IVH) and periventricular leukomalacia (PVL) are major forms of brain injury in extremely preterm infants [4]. Preventive measures, such as antenatal glucocorticoid administration and delayed cord clamping, can lead to a significant reduction of such destructive cerebral injuries [5,6]. More recent studies focused on subtle white matter lesions, which are closely associated with adverse motor and cognitive outcomes [5]. Although white matter lesions are common in moderate and late preterm infants [7], detailed information regarding their type and incidence remains uncovered. This is because these infants undergo MRI less frequently owing to their relatively stable clinical courses.

In this study, we hypothesised that moderate and late preterm infants are also at significant risk of developing white matter lesions associated with clinical indicators of hypoxic-ischaemic stress and transition failure. The study aimed to assess the incidence and clinical risk factors associated with brain lesions, particularly white matter lesions, in moderate and late preterm infants using term-equivalent MRI.

## 2. Materials and Methods

### 2.1. Study Population

Between January 2019 and December 2020, 327 live-born infants at 32+0 to 36+6 weeks of gestation were admitted to the tertiary neonatal intensive care unit at the Japanese Red Cross Aichi Medical Centre Nagoya Daini Hospital (Nagoya, Aichi, Japan) (shown in Figure 1).

At this centre, for infants born before 36 weeks of gestation, infants with congenital brain anomalies, and those who have experienced clinical events leading to hypoxic-ischaemic conditions (such as prolonged resuscitation or the need for mechanical ventilation), MRI scans are scheduled as part of a domestic follow-up protocol.

MRI is typically conducted before discharge and at term-equivalent postmenstrual age (37–40 weeks postmenstrual age) as part of the standard protocol. Infants with chromosomal abnormalities, malformation syndromes, or prenatally diagnosed brain anomalies were excluded. In addition, since the aim of this study was to assess abnormal neuroimaging findings on MRI images taken at term-equivalent postmenstrual age, cases where MRI was performed before 36 weeks of postmenstrual age were excluded to ensure consistency in imaging timing.

### 2.2. MRI Acquisition and Assessment

The scans were conducted during natural sleep. Infants were fed, swaddled, immobilised in a vacuum fixation beanbag, and scanned while sleeping. MRI was performed using a 3-T (Magnetom Skyra, Siemens Healthcare, Erlangen, Germany) or a 1.5-T scanner (Magnetom Avanto or Magnetom Aera, Siemens Healthcare). MRI scans included three-dimensional T1-weighted imaging (1.5T: repetition time 551 msec, echo time 12 msec, slice thickness 5 mm; 3T: repetition time 350 msec, echo time 9 msec, slice thickness 5 mm) and coronal and transverse T2-weighted imaging (1.5T: repetition time 5352 msec, echo time 130 msec, slice thickness 5 mm; 3T: repetition time 4500 msec, echo time 90 msec, slice thickness 5 mm).

In this study, two experienced paediatricians, who had been engaged in MRI studies for 15 and 8 years, respectively, independently assessed scans blinded to the medical history using a combination of sequences of T1- and T2-weighted imaging. MRI findings were systematically categorised into six types: haemorrhage, infarction, deep grey matter lesions, white matter lesions, cerebellar lesions, and other abnormalities. IVH was classified based on Papile’s criteria [8]. Infarctions were defined as focal areas of restricted diffusion or signal abnormalities consistent with acute or chronic ischemic injury. White matter lesions were further categorized into punctate white matter lesions (PWML), cystic PVL, and small focal lesions. PWMLs were defined as small, focal areas of hyperintensity on T1- and/or hypointensity on T2-weighted imaging, typically <5 mm in size and located in the periventricular or subcortical white matter [9]. PVL was defined by cavitation on T2-weighted imaging [10]. A small focal lesion was defined as a well-demarcated lesion with hypointensity and hyperintensity on T1- and T2-weighted imaging, respectively, consistent with fluid-filled characteristics. Deep grey matter and cerebellar lesions were separately recorded. Discrepancies between expert evaluations were resolved by consensus.

### 2.3. Collection of Clinical Data

Relevant clinical data were retrieved from maternal and neonatal records. These data included possible confounders, potentially variables associated with white matter injury or neurodevelopmental outcomes. We used Japanese standards [11] to extract birth weight z-scores for all newborns. Infants were classified as small or large for gestational age when their weight was <10th or >90th percentiles, respectively. Clinical variables are shown in Table 1.

### 2.4. Data Analysis

Statistical analyses were conducted using the IBM SPSS Statistics for Windows, version 29.0 (IBM Corp., Armonk, NY, USA) software. Data are presented as means (standard deviations) unless otherwise stated. Chi-square, independent *t*-tests, and Mann–Whitney U tests (for nonparametric variables) were used for subgroup comparisons. Binary logistic regression analysis was used to identify clinical variables associated with MRI abnormality. *p*-values were not corrected for multiple comparisons due to the exploratory nature of these analyses. Independent variables for the final model were assigned based on clinical relevance (gestational age was used as a mandatory variable) and determined using forward selection.

## 3. Results

### 3.1. Participants

During the study period, 327 infants born between 32+0 and 36+6 weeks of gestation were admitted. Of these, 100 infants did not undergo MRI scans due to the following reasons: gestational age of 36 weeks (*n* = 45), MRI not available by the scheduled date of discharge (*n* = 38), transferred to other hospitals (*n* = 16), and other technical reasons (*n* = 1). Among the 227 infants who underwent MRI scans, 8 with major congenital anomalies, 5 with confirmed chromosomal diseases, 1 with brain abnormalities detected by ultrasound at birth, and 18 infants who underwent MRI scans before 36 weeks postmenstrual age were excluded. The final study cohort consisted of 195 infants. The final cohort had a mean gestational age of 35.0 (1.2) weeks, birth weight of 2100 (379) g, and postmenstrual age of 37.5 (1.2) weeks at MRI, as shown in Table 1 and Figure 1.

Clinical characteristics differed between infants who underwent MRI (*n* = 195) and those who did not (*n* = 100) (Table 1). Infants who underwent MRI had higher rates of caesarean delivery and antenatal corticosteroid use than those who did not. They had a lower gestational age, lower birth weight and its z-score, higher incidence of foetal growth restriction, lower umbilical cord blood pH, and lower 1 min Apgar scores. They were also more likely to require oxygen therapy, non-invasive ventilation, and blood transfusions. Infants who did not undergo MRI were excluded from further analysis.

### 3.2. Incidence of Brain Lesions

Of the 195 infants included in the final study population, 46 (23.6%) had one or more lesions identified on MRI (Table 2). White matter lesions were most common (73.9%) with PWML in 30 cases (65.2%). Of the PWML cases, 22 had fewer than six lesions, while 8 had six or more. Small focal lesions were observed in three cases (6.5%), and one case (2.2%) was diagnosed as PVL. Infarction was present in five cases (10.9%), IVH in four cases (8.7%) (three grade I, one grade II), and deep grey matter lesions in four cases (8.7%). Subependymal cysts and cerebellar lesions were observed in one case each (2.2%).

### 3.3. Dependence of White Matter Lesions on Clinical Variables

Univariate analysis showed that the presence of white matter lesions was associated with vaginal delivery (odds ratio (OR) = 2.915, 95% confidence interval (CI) = 1.241–6.847, *p* = 0.014), male sex (OR = 2.184, 95% CI = 1.007–4.734, *p* = 0.048), larger birth weight (OR = 1.163, 95% CI = 1.048–1.292, *p* = 0.005) and its z-score (OR = 1.640, 95% CI = 1.122–2.397, *p* = 0.011), lower 5 min Apgar scores (OR = 0.710, 95% CI = 0.526–0.958, *p* = 0.025), requirement for tracheal intubation (OR = 2.491, 95% CI = 1.133–5.474, *p* = 0.023), longer duration of intubation (OR = 1.192, 95% CI = 1.019–1.393, *p* = 0.028), and postnatal steroid use (OR = 14.323, 95% CI = 1.442–142.292, *p* = 0.023). In the multivariate model, adjusted for gestational age, white matter lesions were associated with vaginal delivery (*p* = 0.015), larger z-scores of birth weight (*p* = 0.013), and requirement for tracheal intubation (*p* = 0.012) (Table 3).

### 3.4. Dependence of PWML on Clinical Variables

In the univariate analysis, PWML incidence was associated with vaginal delivery (OR = 3.048, 95% CI = 1.254–7.409, *p* = 0.014), premature rupture of membranes (PROM) (OR = 2.824, 95% CI = 1.088–7.329, *p* = 0.033), male sex (OR = 2.382, 95% CI = 1.044–5.435, *p* = 0.039), larger birth weight (OR = 1.172, 95% CI = 1.050–1.309, *p* = 0.005) and its z-score (OR = 1.749, 95% CI = 1.167–2.621, *p* = 0.007), lower 5 min Apgar scores (OR = 0.697, 95% CI = 0.513–0.945, *p* = 0.020), the requirement for tracheal intubation (OR = 2.779, 95% CI = 1.194–6.469, *p* = 0.018), and longer duration of intubation (OR = 1.191, 95% CI = 1.015–1.396, *p* = 0.032). In the multivariate model adjusted for gestational age, PWML was associated with larger z-scores of birth weight (*p* = 0.005) and requirement for tracheal intubation (*p* = 0.013) (Table 4).

See Online Appendix A for alternative analyses showing consistent relationships between PWML of six or more lesions and PROM, larger birth weight, lower 5 min Apgar scores, need of resuscitation, requirement for tracheal intubation, and later postnatal age at scan.

### 3.5. Dependence of Grey Matter Lesions on Clinical Variables

In the univariate analysis, the presence of grey matter lesions was associated with foetal distress (OR = 13.900, 95% CI = 1.767–109.314, *p* = 0.012), lower 1 (OR = 0.592, 95% CI = 0.411–0.854, *p* = 0.005) and 5 min Apgar scores (OR = 0.504, 95% CI = 0.304–0.835, *p* = 0.008), lower umbilical cord pH (OR = 0.280, 95% CI = 0.121–0.651, *p* = 0.003), lower base excess (OR = 0.726, 95% CI = 0.588–0.897, *p* = 0.003), longer duration of intubation (OR = 1.455, 95% CI = 1.028–2.060, *p* = 0.034), inotrope use (OR = 86.400, 95% CI = 7.588–983.835, *p* < 0.001), and postnatal steroid administration (OR = 49.333, 95% CI = 2.461–989.130, *p* = 0.011) (Online Appendix A). Due to the limited number of outcome events (fewer than 10), a multivariable logistic regression analysis was not performed to avoid overfitting and unreliable estimates.

### 3.6. Dependence of Any Brain Lesions on Clinical Variables

Univariate analysis demonstrated that the incidence of any brain lesions (*n* = 46) was associated with vaginal birth (OR = 2.949, 95% CI = 1.366–6.370, *p* = 0.006), male sex (OR = 2.032, 95% CI = 1.029–4.011, *p* = 0.041), larger birth weight (OR = 1.105, 95% CI = 1.010–1.209, *p* = 0.029) and its z-score (OR = 1.465, 95% CI = 1.052–2.040, *p* = 0.024), lower 5 min Apgar scores (OR = 0.731, 95% CI = 0.560–0.955, *p* = 0.022), lower cord base excess (OR = 0.877, 95% CI = 0.788–0.976, *p* = 0.016), the requirement for tracheal intubation (OR = 2.462, 95% CI = 1.228–4.936, *p* = 0.011), longer duration of intubation (OR = 1.185, 95% CI = 1.022–1.373, *p* = 0.024), inotrope use (OR = 4.320, 95% CI = 1.253–14.890, *p* = 0.020), and postnatal steroid administration (OR = 14.095, 95% CI = 1.534–129.512, *p* = 0.019). In the multivariate model, adjusted for sex and gestational age, brain lesions were associated with vaginal delivery (*p* = 0.014), larger z-scores of birth weight (*p* = 0.043), and requirement for intubation (*p* = 0.008) (Online Appendix A).

## 4. Discussion

In a large cohort of moderate and late preterm infants, 23.6% exhibited brain abnormalities on MRI, with white matter lesions being the most prevalent. PWML accounted for 65.2% of the total brain abnormalities. PWML incidence in moderate and late preterm infants (15.4%) was comparable to that observed in extremely preterm newborns (18.8%) [12]. Clinical risk factors associated with PWML included the need for intubation, intrauterine growth, and delivery mode. Further follow-up studies are necessary to elucidate the association between brain lesions and neurodevelopmental outcomes.

In very preterm infants (<32 weeks), severe forms of brain injury, such as PVL, IVH, and periventricular haemorrhagic infarction are linked to impaired neurodevelopment [13]. Advances in neonatal care have decreased the incidence of cystic PVL and severe IVH [14], shifting attention to subtler white matter injuries such as PWML and diffuse excessive high-signal intensity [15]. However, while extensively studied in very preterm [12,16], in moderate and late preterm infants, such abnormalities remain underreported [17].

Our study demonstrated that white matter lesions, especially PWML, are major forms of brain injury even in moderate and late preterm infants, consistent with Boswinkel et al. [18]. Our study further demonstrated that large body size, vaginal delivery, and transition failure requiring tracheal intubation were independent variables associated with increased incidence of white matter lesions and PWML. Miller et al. reported that PWML with six or more lesions is associated with poor neurodevelopmental outcomes [15]. However, our study demonstrated that independent variables of PWML remain relevant regardless of the number of lesions, confirming the importance of elucidating the specific long-term neurodevelopmental outcomes associated with PWML.

Previous studies also identified the association between mechanical ventilation and PWML in very preterm infants [16]. Despite the short duration of mechanical ventilation in our study (mean 1.4 ± 2.4 days), a longer intubation period was significantly associated with PWML (OR = 1.191, 95% CI = 1.015–1.396, *p* = 0.032). Mechanical ventilation is associated with an increased risk of developing cystic PVL, in part via hypocapnia [19]. However, the relationship between mechanical ventilation, hypocapnia, and PWML development remains uncertain. At our institution, close end-tidal CO_2_ monitoring during ventilation minimises the incidence of prolonged hypocapnia. It is plausible, therefore, that the requirement for respiratory support during the neonatal transitional period may be responsible for PWML development.

It was notable that large body size was associated with PWML. Wagenaar et al. reported a similar relationship between body size and the incidence of PWML [12]. They speculated that larger infants undergo MRI earlier than smaller ones, meaning that the scans are performed before tissue repair and consolidation occur, which may explain why PWML is more frequently detected in larger infants. In our study, a significant negative correlation was observed between larger z-scores and the postnatal age at MRI (r = −0.303, *p* < 0.001), suggesting the presence of a similar phenomenon to that reported by Wagenaar et al. [12]. There are several possible explanations for the relationship between larger birth weight z-scores and the incidence of PWML. One possibility is prenatal exposure to maternal cortisol, which enhances foetal growth and maturation [20] but contributes to white matter injuries by inhibiting pre-myelinating oligodendrocytes [21]. Another explanation is that larger z-scores of birth weight are associated with the incidence of respiratory morbidities [22]. Additionally, gestational diabetes, a known cause of foetal overgrowth, has been linked to a higher risk of respiratory distress syndrome, particularly in late preterm and term infants [23]. Although our study did not assess the presence of gestational diabetes, it is possible that larger body size may contribute to an increased incidence of PWML through the development of relatively more severe respiratory morbidities. Future studies are needed to validate these hypotheses.

Inflammation plays a key role in white matter injury development [24] and is also linked with preterm delivery [25]. In this study, PROM and vaginal delivery were associated with PWML. As the development of cystic PVL is explained by multifactorial mechanisms, including hypocapnia, infection, inflammation, and hypoperfusion, PWML development may also be explained by multiple upstream events and conditions [25].

Basal ganglia injuries were associated with foetal distress, inotrope use, and postnatal steroids, indicating difficulties in transition after birth. These findings align with the established understanding that severe hypoxic-ischaemic events or circulatory failure predominantly affect metabolically active brain regions, such as the basal ganglia [26]. Importantly, our results underscore that even in moderate and late preterm infants, basal ganglia injuries can occur in perinatal compromise. Proactive neuroimaging in this population may facilitate the timely identification of such injuries, enabling targeted interventions to mitigate potential long-term neurodevelopmental impairments.

In contrast to Boswinkel et al. [18], who reported a 73.2% incidence of brain lesions in moderate and late preterm infants, our study found a lower frequency of 23.6% of cases with MRI abnormalities. This discrepancy may be attributed to differences in lesion evaluation, as their study considered maturation-related changes and ventricular enlargement in addition to overt abnormalities. Additionally, unlike Boswinkel et al., who studied infants born between 32 and 35 weeks [18], our cohort included infants born at 36 weeks of gestation (26.7% of total), which may have contributed to the differences in findings. Despite these variations, both studies consistently found that mild lesions, such as white matter lesions, were the most common type of brain injury. Furthermore, our study identified clinical factors associated with white matter lesions, including PWML, using logistic regression analysis. Importantly, our findings indicate that white matter lesions, including PWML, can occur across all gestational ages within the moderate and late preterm group. This suggests PWML could develop beyond the peak period of oligodendrocyte progenitor cells (25–34 weeks) [21]. While further validation across diverse populations and treatment settings is necessary, these findings highlight the need for attention to PWML risk across a broader range of gestational ages.

This study had a few limitations. First, although we investigated several important factors, we were unable to account for all potential confounders, such as inflammatory markers and prenatal blood glucose regulation. Therefore, it is possible that certain clinical management factors influencing brain lesion development were overlooked. Furthermore, due to the exploratory nature of this study, we did not perform multiple comparisons in our statistical analysis. Consequently, we have carefully interpreted the results, considering *p*-values between 0.01 and 0.05 as suggestive rather than definitive. This limitation should be acknowledged when interpreting the findings of this study. Second, infants with more severe clinical conditions were more likely to undergo MRI. Nonetheless, postnatal clinical factors, including the need for resuscitation, duration of mechanical ventilation, and inotrope use, did not differ significantly between infants who underwent MRI and those who did not. Third, this study lacked neurodevelopmental correlation with abnormal imaging findings, which remains a critical limitation. We are currently conducting follow-up research to evaluate the neurodevelopmental outcomes of the study cohort. While most of the reported brain lesions were mild and some may have been incidental findings, their long-term clinical significance remains unclear. Given the potential impact of even subtle white matter abnormalities on later neurodevelopment, comprehensive follow-up studies are essential to determine their prognostic value and to guide early interventions.

## 5. Conclusions

In moderate and late preterm infants, white matter injury, particularly PWML, was observed in 15.4% of the total cohort of moderate and late preterm infants. PWML is associated with the need for intubation, larger body size, and vaginal delivery, highlighting the importance of antenatal growth, respiratory transition, and inflammation in understanding the mechanisms of subtle brain injury in moderate and late preterm infants. This relatively high prevalence may underscore the need for increased monitoring of brain health across a broader range of gestational ages, including moderate and late preterm infants. Future research should focus on detailed clinical care measures and neurodevelopmental assessments to help identify modifiable risk factors for brain injury and related outcomes in moderate and late preterm infants.

## Figures and Tables

**Figure 1 diagnostics-15-00881-f001:**
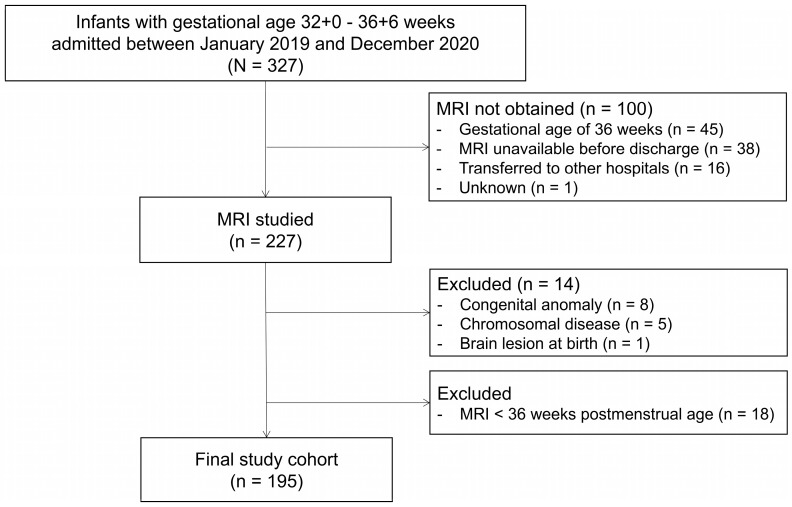
Flowchart of the study population of newborn infants with MRI evaluation.

**Table 1 diagnostics-15-00881-t001:** Comparisons of clinical characteristics of newborn infants with and without MRI evaluation.

Variables	Evaluated (*n* = 195)	Not Evaluated (*n* = 100)	*p*
Maternal and antenatal variables			
Parity (multipara)	86 (44.1)	42 (42.0)	0.730
Multiple pregnancy	61 (31.3)	31 (31.0)	0.961
Caesarean section	159 (81.5)	71 (71.0)	0.039
Premature rupture of the membrane	11 (5.6)	2 (2.0)	0.231
Antenatal steroids	73 (37.4)	20 (20.2)	0.003
Clinical chorioamnionitis	9 (4.8)	1 (1.0)	0.172
Outborn	9 (4.6)	9 (9.0)	0.136
Variables at birth			
Male sex	97 (49.7)	59 (59.0)	0.132
Gestational age (week)	35.0 (1.2)	35.7 (0.9)	<0.001
32 weeks	11 (5.6)	1 (1.0)	
33 weeks	31 (15.9)	3 (3.0)	
34 weeks	54 (27.7)	6 (6.0)	
35 weeks	47 (24.1)	45 (45.0)	
36 weeks	52 (26.7)	45 (45.0)	
Body weight at birth (g)	2100 (379)	2379 (348)	<0.001
Body weight z score	−0.50 (1.07)	−0.07 (0.90)	<0.001
Intrauterine growth restriction	23 (11.8)	2 (2.0)	0.004
Apgar score			
1 min.	8 [8–8]	8 [8–8]	0.008
5 min.	9 [8–9]	9 [8–9]	0.058
Cord blood			
pH	7.312 (0.076)	7.326 (0.038)	0.021
Base excess (mmol/L)	−2.01 (3.35)	−1.92 (1.52)	0.291
Need of resuscitation	94 (48.2)	41 (41.0)	0.240
Variables after admission			
Oxygen supplement	178 (91.3)	80 (81.6)	0.016
Non-invasive mechanical ventilation	148 (75.4)	51 (51.0)	<0.001
Need for intubation	99 (50.8)	46 (46.5)	0.485
Duration of mechanical ventilation	1.4 (2.4)	1.0 (1.4)	0.070
Any treatment for PDA	7 (3.6)	2 (2.0)	0.723
Inotropes use	10 (5.1)	1 (1.0)	0.065
Transfusion	10 (5.1)	0 (0.0)	0.018
Postnatal steroids	5 (2.6)	0 (0.0)	0.171
MRI scan			
Postmenstrual age at scan (weeks)	37.5 (1.2)	N/A	
Postnatal age at scan (days)	18.0 (10.7)	N/A	

Values are shown as the number (%), mean ± standard deviation or median [interquartile range]. PDA, patent ductus arteriosus; MRI, magnetic resonance imaging.

**Table 2 diagnostics-15-00881-t002:** Incidence of brain lesions.

Brain Lesions	*n* = 46
Haemorrhages	
IVH	
Total	4 (8.7)
Grade I	3
Grade II	1
Infarction	5 (10.9)
Deep grey matter	
Small focal lesion	4 (8.7)
White matter	
Punctate white matter lesions (PWML)	30 (65.2)
<6 PWML	22
≥6 PWML	8
Periventricular leukomalacia	1 (2.2)
Small focal lesion	3 (6.5)
Miscellaneous	
Subependymal cyst	1 (2.2)
Cerebellum	1 (2.2)

Values are shown as the number (%). IVH, intraventricular haemorrhage.

**Table 3 diagnostics-15-00881-t003:** Dependence of white matter lesions on clinical variables.

	No Lesion *n* = 149	White Matter Lesion *n* = 34	OR	95% CI		*p*
				Lower	Upper	
**Univariate analysis**						
Maternal and antenatal variables						
Vaginal delivery	21 (14.1)	11 (32.4)	2.915	1.241	6.847	0.014
PROM	10 (6.7)	1 (2.9)	0.421	0.052	3.407	0.418
Antenatal steroids	53 (35.6)	14 (41.2)	1.268	0.592	2.714	0.541
Clinical chorioamnionitis	5 (3.4)	3 (9.4)	2.917	0.660	12.894	0.158
Threatened preterm labour	79 (53.0)	19 (55.9)	1.122	0.530	2.375	0.763
Foetal distress	10 (6.7)	1 (2.9)	0.421	0.052	3.407	0.418
Placenta praevia	6 (4.0)	0 (0.0)	N/A			N/A
Hypertensive disorders in pregnancy	20 (13.4)	1 (2.9)	0.195	0.025	1.510	0.118
Variables at birth						
Male sex	68 (45.6)	22 (64.7)	2.184	1.007	4.734	0.048
Gestational age (week)	35.0 (1.3)	35.1 (1.1)	1.076	0.793	1.459	0.639
Body weight at birth (per 100g)	2067 (371)	2277 (372)	1.163	1.048	1.292	0.005
Body weight z score	−0.60 (1.10)	−0.06 (0.92)	1.640	1.122	2.397	0.011
Apgar score						
1 min.	8 [8–8]	8 [8–8]	0.882	0.712	1.093	0.251
5 min.	9 [8–9]	9 [8–9]	0.710	0.526	0.958	0.025
Cord blood						
pH (per 0.1)	7.318 (0.051)	7.308 (0.087)	0.766	0.433	1.355	0.359
Base excess (mmol/L)	−1.7 (2.6)	−2.8 (4.2)	0.891	0.790	1.005	0.061
Need of resuscitation	66 (44.3)	19 (55.9)	1.593	0.752	3.373	0.224
Need for intubation	68 (45.6)	23 (67.6)	2.491	1.133	5.474	0.023
Duration of intubation	1.2 (1.8)	2.4 (3.9)	1.192	1.019	1.393	0.028
Any treatment for PDA	7 (4.7)	0 (0.0)	N/A			N/A
Inotropes use	5 (3.4)	3 (8.8)	2.787	0.633	12.281	0.176
Transfusion	7 (4.7)	3 (8.8)	1.963	0.481	8.018	0.347
Postnatal steroid	1 (0.7)	3 (8.8)	14.323	1.442	142.292	0.023
MRI scan						
Postmenstrual age at scan (weeks)	37.6 (1.3)	37.2 (1.1)	0.716	0.477	1.074	0.107
Postnatal age at scan (days)	18.5 (10.9)	14.9 (10.1)	0.962	0.921	1.005	0.083
**Multivariate analysis**						
Gestational age (week)			1.184	0.855	1.639	0.309
Vaginal delivery			3.102	1.250	7.696	0.015
Body weight at birth (z score)			1.702	1.118	2.591	0.013
Need for intubation			2.948	1.269	6.850	0.012

Abbreviation: CI, confidence interval; OR, odds ratio; PROM, premature rupture of the membrane; PDA, patent ductus arteriosus; MRI, magnetic resonance imaging.

**Table 4 diagnostics-15-00881-t004:** Dependence of punctate white matter lesions on clinical variables.

	OR	95% CI		*p*
		Lower	Upper	
**Univariate analysis**				
Maternal and antenatal variables				
Vaginal delivery	3.048	1.254	7.409	0.014
PROM	2.824	1.088	7.329	0.033
Antenatal steroids	1.385	0.625	3.071	0.423
Clinical chorioamnionitis	3.384	0.760	15.063	0.110
Threatened preterm labour	1.013	0.461	2.223	0.975
Foetal distress	N/A			N/A
Placenta praevia	N/A			N/A
Hypertensive disorders in pregnancy	0.222	0.029	1.725	0.150
Variables at birth				
Male sex	2.382	1.044	5.435	0.039
Gestational age (week)	1.040	0.757	1.430	0.808
Body weight at birth (per 100g)	1.172	1.050	1.309	0.005
Body weight z score	1.749	1.167	2.621	0.007
Apgar score				
1 min.	1.056	0.482	2.314	0.892
5 min.	0.697	0.513	0.945	0.020
Cord blood				
pH (per 0.1)	1.056	0.482	2.314	0.892
Base excess (mmol/L)	0.910	0.784	1.057	0.217
Need of resuscitation	1.645	0.745	3.628	0.218
Need for intubation	2.779	1.194	6.469	0.018
Duration of intubation	1.191	1.015	1.396	0.032
Any treatment for PDA	N/A			N/A
Inotropes use	2.057	0.380	11.138	0.403
Transfusion	1.449	0.286	7.343	0.654
Postnatal steroid	10.571	0.927	120.585	0.058
MRI scan				
Postmenstrual age at scan (weeks)	0.659	0.420	1.034	0.069
Postnatal age at scan (days)	0.960	0.917	1.006	0.087
**Multivariate analysis**				
Gestational age (week)	1.160	0.827	1.628	0.390
Body weight at birth (z score)	1.890	1.215	2.940	0.005
Need for intubation	3.094	1.274	7.513	0.013

Abbreviation: CI, confidence interval; OR, odds ratio; PROM, premature rupture of the membrane; PDA, patent ductus arteriosus; MRI, magnetic resonance imaging.

## Data Availability

The data supporting the findings of this study are not publicly available because they contain information that could compromise participant privacy. However, the data are available from the corresponding author (K.T.) upon reasonable request.

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
