# Peer review of "Incidence and Risk Factors of White Matter Lesions in Moderate and Late Preterm Infants"

_diagnostics, 2025, doi:10.3390/diagnostics15070881_

Round 1
Reviewer 1 Report
Comments and Suggestions for Authors
The manuscript by Kentaro Ueda et al. entitled “Incidence and Risk Factors of White Matter Lesions in Moderate and Late Preterm Infants” focuses on an interesting topic in perinatal medicine. Particularly, authors attempted to evaluate the incidence and clinical risk factors associated with brain lesions (particularly white matter lesions), in moderate and late preterm infants using term-equivalent MRI.
The study is overall well designed and well written, with interesting results.
However, in the "Material and Methods" section I recommend better specifying the decision to divide the population study into the two study groups ("evaluated MRI" and "no MRI") because it is missing in the manuscript text (it is only evident from the study design figure).
In this regard I advice you to justify the reason why you had excluded 100 premature newborn from MRI assessment, also having significantly different clinical characteristics as shown by Table 1 (i.e. birth weight, z score, non-invasive ventilation).
Did the 100 excluded have a gestational age between 36 and 36+6? Please specify
Author Response
Reviewer Comments and Author Response:
The authors are grateful to the reviewers for their insightful comments on our manuscript. We have revised the manuscript in light of these valuable comments.
Reviewer 1:
Comments to the Author
The manuscript by Kentaro Ueda et al. entitled “Incidence and Risk Factors of White Matter Lesions in Moderate and Late Preterm Infants” focuses on an interesting topic in perinatal medicine. Particularly, authors attempted to evaluate the incidence and clinical risk factors associated with brain lesions(particularly white matter lesions), in moderate and late preterm infants using term-equivalent MRI.
The study is overall well designed and well written, with interesting results. However, in the "Material and Methods" section I recommend better specifying the decision to divide the population study into the two study groups ("evaluated MRI" and "no MRI") because it is missing in the manuscript text (it is only evident from the study design figure).In this regard I advice you to justify the reason why you had excluded 100 premature newborn from MRI assessment, also having significantly different clinical characteristics as shown by Table 1 (i.e. birth weight, z score, non-invasive ventilation).
Did the 100 excluded have a gestational age between 36 and36+6? Please specify
- Response to the Reviewer: We appreciate the precise summary of our findings and the constructive comments provided by the reviewer. We agree that information regarding the decision criteria for MRI assessment is essential in considering potential selection bias in our study. Because this study was a retrospective observational study, we had described in the Method section of the original manuscript that “At this centre, for infants born before 36 weeks of gestation, infants with congenital brain anomalies, and those who have experienced clinical events leading to hypoxic-ischaemic conditions, such as prolonged resuscitation or the need for mechanical ventilation, MRI scans are scheduled as part of a domestic follow-up protocol.”. This means that those with MRI are inevitably more immature and exposed to hypoxic-ischaemic stress compared to their peers.
To further clarify the details of the 100 excluded infants, we have added the following information to Figure 1 and section 3.1, Participants in the Results:
"Of these, 100 infants did not undergo MRI scans due to the following reasons: gestational age of 36 weeks (n = 45), MRI not available by the scheduled date of discharge (n = 38), transferred to other hospitals (n = 16), and other technical reasons (n = 1). Among the 227 infants who underwent MRI scans, eight with major congenital anomalies, five with confirmed chromosomal diseases, one with brain abnormalities detected by ultrasound at birth, and 18 infants who underwent MRI scans before 36 weeks postmenstrual age were excluded. The final study cohort consisted of 195 infants."
We believe this added information clarifies the selection criteria for our study and addresses the reviewer's concerns regarding potential selection bias.
Reviewer 2 Report
Comments and Suggestions for Authors
This paper is well-structured and presents a clear research question, methodology, and results. However, with some revisions to improve clarity, readability, and scientific rigor, the paper will be a valuable contribution to the field of neonatal neurology
Q1: line 11: "Incidences of white matter lesions, which are associated with developmental impairments in very preterm infants, remain underreported in this population." Consider to rephrase as: Change "Incidences" to "Incidence", and remain to remains for grammatical correctness.
Q2: line 24-25: "Future research should focus on detailed clinical care measures and neurodevelopmental assessments may help identify modifiable risk factors for brain injury." Consider to rephrase as:: "Future research should focus on detailed clinical care measures and neurodevelopmental assessments to identify modifiable risk factors for brain injury."
Q3: line 127: "Infants without MRI studies were not discussed further." Consider to rephrase as: "Infants who did not undergo MRI were excluded from further analysis."
Q4: line 67: Line 67 "cases where MRI was performed before 36 weeks of gestation were also excluded" Consider to rephrase as: "cases where MRI was performed before 36 weeks of postmenstrual age were excluded to ensure consistency in imaging timing."
Q5: Although the statistical analysis is appropriate, however, the authors should clarify why they did not correct for multiple comparisons, as this could increase the risk of Type I errors.
Q6: Statistical analysis: This may raise concerns about false positive. A Bonferroni correction or a statement justifying why correction wasn’t applied should be included
Q7: The description of MRI protocols is clear, but there was no mention of inter-rater reliability.
- Were scans evaluated independently by two experts
- Was there a consensus protocol for discrepancies?
- You may consider adding” Two independent radiologists assessed scans. Discrepancies were resolved by consensus
Q8: Potential unmeasured confounders (such as maternal infections, inflammatory markers) may influence results
Q9: My concerns regarding your results are: the association between larger birth weight z-scores and PWML could be due to
- Higher metabolic demand in larger infants
- Differences in timing of MRI scans
Q10: Does your result add novel findings, such as a higher prevalence of PWML in a different gestation range?
Q11: The discussion includes a comparison with previous studies, but some references (especially [18]) should be analyzed more critically to highlight differences of novel findings.
Q12: Limitations: The study design is robust, with a clear prospective observational approach. The limitations section is well-written but could be expanded. For example, the authors could discuss the potential impact of selection bias, as infants with more severe clinical courses were more likely to undergo MRI. Additionally, the lack of neurodevelopmental follow-up is a significant limitation. This should be explicitly acknowledged in the discussion, along with a plan for future research to address this gap.
Q13: The identification of vaginal delivery, larger birth weight z-scores, and intubation as risk factors is well-supported by the data. However, the discussion could explore potential mechanisms in more depth. For example, why might larger birth weight z-scores increase the risk of PWML? Is it related to maternal factors, such as gestational diabetes, or neonatal factors, such as respiratory distress
Author Response
Reviewer Comments and Author Response:
The authors are grateful to the reviewers for their insightful comments on our manuscript. We have revised the manuscript in light of these valuable comments.
Reviewer 2:
This paper is well-structured and presents a clear research question, methodology, and results. However, with some revisions to improve clarity, readability, and scientific rigor, the paper will be a valuable contribution to the field of neonatal neurology
Q1: line 11: "Incidences of white matter lesions, which are associated with developmental impairments in very preterm infants, remain underreported in this population." Consider to rephrase as: Change "Incidences" to "Incidence", and remain to remains for grammatical correctness.
Q2: line 24-25: "Future research should focus on detailed clinical care measures and neurodevelopmental assessments may help identify modifiable risk factors for brain injury." Consider to rephrase as:: "Future research should focus on detailed clinical care measures and neurodevelopmental assessments to identify modifiable risk factors for brain injury."
Q3: line 127: "Infants without MRI studies were not discussed further." Consider to rephrase as: "Infants who did not undergo MRI were excluded from further analysis."
Q4: line 67: Line 67 "cases where MRI was performed before 36weeks of gestation were also excluded" Consider to rephrase as:"cases where MRI was performed before 36 weeks of postmenstrual age were excluded to ensure consistency in imaging timing."
- Response to the Reviewer: We appreciate the editor’s valuable feedback.
Q1: We have revised the sentence as suggested, changing "Incidences" to "Incidence" and "remain" to "remains" for grammatical accuracy.
Q2: We have modified the sentence as recommended to improve clarity.
Q3: We have revised the sentence to "Infants who did not undergo MRI were excluded from further analysis," as suggested.
Q4: We have reworded the sentence to "Cases where MRI was performed before 36 weeks of postmenstrual age were excluded to ensure consistency in imaging timing," as recommended.
Thank you for your thoughtful suggestions, which helped improve the clarity and readability of our manuscript.
Q5: Although the statistical analysis is appropriate, however, the authors should clarify why they did not correct for multiple comparisons, as this could increase the risk of Type I errors.
Q6: Statistical analysis: This may raise concerns about false positive. A Bonferroni correction or a statement justifying why correction wasn’t applied should be included
- Response to the Reviewer: We appreciate the reviewer’s insightful comment. As this study was exploratory in nature, we aimed to generate hypotheses rather than test a pre-specified one. Given this, we did not apply strict multiple comparison corrections, as doing so could significantly reduce statistical power and potentially overlook meaningful findings. Instead, we have carefully interpreted results with p-values between 0.01 and 0.05 as requiring further validation in future studies.
To address the reviewer's concern, we have explicitly acknowledged this limitation in the manuscript. Specifically, we have added the following to the Limitations section:
"Furthermore, due to the exploratory nature of this study, we did not perform multiple comparisons in our statistical analysis. Consequently, we have carefully interpreted the results, considering p-values between 0.01 and 0.05 as suggestive rather than definitive. This limitation should be acknowledged when interpreting the findings of this study."
We believe this addition provides greater transparency regarding our statistical approach and appropriately contextualizes the findings within the limitations of an exploratory study.
Q7: The description of MRI protocols is clear, but there was no mention of inter-rater reliability.
- Were scans evaluated independently by two experts
- Was there a consensus protocol for discrepancies?
- You may consider adding” Two independent radiologists assessed scans. Discrepancies were resolved by consensus
- Response to the Reviewer: Thank you for your important comment regarding MRI evaluation methods. The scans were independently assessed by two experienced paediatricians, who had been engaged in MRI studies for 15 and eight years. In cases of discrepancies, both experts reviewed the images together to reach a consensus. We have added this information to the Method
Q8: Potential unmeasured confounders (such as maternal infections, inflammatory markers) may influence results
- Response to the Reviewer: Thank you for your insightful comment. As you pointed out, our study may be influenced by potential unmeasured confounders, such as maternal infections or inflammatory markers. We have acknowledged this limitation in the Discussion section by adding the following:
"First, although we investigated several important factors, we were unable to account for all potential confounders, such as inflammatory markers and prenatal blood glucose regulation. Therefore, it is possible that certain clinical management factors influencing brain lesion development were overlooked."
Due to the constraints of the revision period, we were unable to include additional analyses on these factors, but we recognize their potential impact and have highlighted this as an important area for future research.
Q9: My concerns regarding your results are: the association between larger birth weight z-scores and PWML could be due to
- Higher metabolic demand in larger infants
- Differences in timing of MRI scans
- Response to the Reviewer: Thank you for your insightful comments. Given that infants with larger birth weight z-scores are at higher risk for respiratory morbidities, it is possible that increased metabolic demand resulting from respiratory distress may contribute to the observed association with PWML. Regarding the timing of MRI scans, we have analysed the correlation between birth weight z-scores and postnatal age at MRI acquisition, finding a negative correlation (r = -0.303, p < 0.001). This suggests that infants with higher birth weight z-scores tended to undergo MRI at an earlier postnatal age, which could potentially influence our findings. To address this point, we have added the following to the Discussion section:
"In our study, a significant negative correlation was observed between larger z-scores and the postnatal age at MRI (r = -0.303, p < 0.001), suggesting the presence of a similar phenomenon to that reported by Wagenaar et al."
We believe this addition provides further context for interpreting our results and acknowledges the potential influence of MRI timing on the observed association between birth weight z-scores and PWML.
Q10: Does your result add novel findings, such as a higher prevalence of PWML in a different gestation range?
- Response to the Reviewer: Yes, as you correctly noted, previous studies on PWML have primarily focused on infants born before 28 weeks of gestation. Our study provides novel findings by demonstrating that PWML is observed at a similar prevalence in moderate and late preterm infants. We have revised the Conclusion section to emphasize this point further.
Q11: The discussion includes a comparison with previous studies, but some references (especially [18]) should be analyzed more critically to highlight differences of novel findings.
- Response to the Reviewer: Thank you for your valuable comment. While the study by Boswinkel et al. similarly investigates brain abnormalities in moderate and late preterm infants, our study differs in two key aspects. First, our study included infants born at 36 weeks of gestation, who comprised 26.7% of our study population. Second, our study examined various associated risk factors for brain abnormalities, providing a more comprehensive understanding of their aetiology. To emphasize these differences and highlight the novel aspects of our findings, we have revised the Discussion section as follows:
"In contrast to Boswinkel et al., who reported a 73.2% incidence of brain lesions in moderate and late preterm infants, our study found a lower frequency of 23.6% of cases with MRI abnormalities. This discrepancy may be attributed to differences in lesion evaluation, as their study considered maturation-related changes and ventricular enlargement in addition to overt abnormalities. Additionally, unlike Boswinkel et al., who studied infants born between 32-35 weeks, our cohort included infants born at 36 weeks of gestation (26.7% of total), which may have contributed to the differences in findings. Despite these variations, both studies consistently found that mild lesions, such as white matter lesions, were the most common type of brain injury. Furthermore, our study identified clinical factors associated with white matter lesions, including PWML, using logistic regression analysis. Importantly, our findings indicate that white matter lesions, including PWML, can occur across all gestational ages within the moderate and late preterm group."
We believe these revisions clarify the distinctions between our study and that of Boswinkel et al., and underscore the unique contributions of our findings.
Q12: Limitations: The study design is robust, with a clear prospective observational approach. The limitations section is well-written but could be expanded. For example, the authors could discuss the potential impact of selection bias, as infants with more severe clinical courses were more likely to undergo MRI. Additionally, the lack of neurodevelopmental follow-up is a significant limitation. This should be explicitly acknowledged in the discussion, along with a plan for future research to address this gap.
- Response to the Reviewer: We appreciate your insightful suggestion regarding the potential for selection bias and the lack of neurodevelopmental follow-up in our study. We agree that these are important considerations. To address these points, we have revised the Limitations section of our manuscript as follows:
"Second, as infants with more severe clinical conditions were more likely to undergo MRI, selection bias may be present. Nonetheless, postnatal clinical factors, including the need for resuscitation, duration of mechanical ventilation, and inotrope use, did not differ significantly between infants who underwent MRI and those who did not. Third, this study lacked neurodevelopmental correlation with abnormal imaging findings, which remains a critical limitation. We are currently conducting follow-up research to evaluate the neurodevelopmental outcomes of the study cohort. While most of the reported brain lesions were mild and some may have been incidental findings, their long-term clinical significance remains unclear. Given the potential impact of even subtle white matter abnormalities on later neurodevelopment, comprehensive follow-up studies are essential to determine their prognostic value and to guide early interventions."
This revised section now explicitly acknowledges the potential for selection bias, despite the lack of significant differences in postnatal clinical factors between the groups. Furthermore, it clearly states the limitations of the current study due to the absence of neurodevelopmental follow-up and emphasizes the importance of ongoing and future research to address this critical gap in our knowledge.
Q13: The identification of vaginal delivery, larger birth weight z-scores, and intubation as risk factors is well-supported by the data. However, the discussion could explore potential mechanisms in more depth. For example, why might larger birthweight z-scores increase the risk of PWML? Is it related to maternal factors, such as gestational diabetes, or neonatal factors, such as respiratory distress.
- Response to the Reviewer: We appreciate the reviewer’s insightful comment regarding the association between larger birth weight z-scores and PWML. While maternal factors such as gestational diabetes and maternal nutrition may contribute to foetal overgrowth, our study was unable to assess these variables. However, we acknowledge the potential link between gestational diabetes, foetal overgrowth, and respiratory distress, which could contribute to the observed association with PWML. To address this point, we have expanded the Discussion section with the following:
"Additionally, gestational diabetes, a known cause of foetal overgrowth, has been linked to a higher risk of respiratory distress syndrome, particularly in late preterm and term infants. Although our study did not assess the presence of gestational diabetes, it is possible that larger body size may contribute to an increased incidence of PWML through the development of relatively more severe respiratory morbidities."
This addition highlights the potential role of gestational diabetes and respiratory distress as contributing factors to the observed association, even though our study did not directly assess these variables.
Reviewer 3 Report
Comments and Suggestions for Authors
General Comments
This is a descriptive study that purports to show that in moderate and late preterm infants, white matter injury (PWML), was observed in 17.4% of the total cohort. PWML was associated with the need for intubation, larger body size, and vaginal delivery. However, there are multiple omissions in the data as there is no follow-up data on this group of infants and there is no control group for comparison. Moreover, although the larger infants appeared to be at higher risk, the incidence of gestational diabetes or diabetes in the mothers is not indicated nor is the incidence of hypoglycemia.
Specific Comments
- Line 40: Surfactant replacement has not been shown to affect neurological outcomes/ IVH.
- It is surprising that they have detected this amount of injury in relatively late gestation infants.
Author Response
Reviewer Comments and Author Response:
The authors are grateful to the reviewers for their insightful comments on our manuscript. We have revised the manuscript in light of these valuable comments.
Reviewer 3:
This is a descriptive study that purports to show that in moderate and late preterm infants, white matter injury (PWML), was observed in 17.4% of the total cohort. PWML was associated with the need for intubation, larger body size, and vaginal delivery. However, there are multiple omissions in the data as there is no follow-up data on this group of infants and there is no control group for comparison. Moreover, although the larger infants appeared to be at higher risk, the incidence of gestational diabetes or diabetes in the mothers is not indicated nor is the incidence of hypoglycemia.
- Response to the Reviewer: Thank you for summarizing our study and providing constructive feedback. We acknowledge that the lack of follow-up data is a limitation of our study. However, conducting long-term follow-up for all moderate and late preterm infants presents a significant healthcare and economic burden. We believe that our study contributes to identifying high-risk groups, which can help prioritize follow-up efforts more efficiently. Currently, we are conducting research to investigate how MRI abnormalities affect long-term outcomes.
We have revised the Limitations section of our manuscript as follows:
"Third, this study lacked neurodevelopmental correlation with abnormal imaging findings, which remains a critical limitation. We are currently conducting follow-up research to evaluate the neurodevelopmental outcomes of the study cohort. While most of the reported brain lesions were mild and some may have been incidental findings, their long-term clinical significance remains unclear. Given the potential impact of even subtle white matter abnormalities on later neurodevelopment, comprehensive follow-up studies are essential to determine their prognostic value and to guide early interventions."
Additionally, as you correctly pointed out, maternal factors such as gestational diabetes and neonatal factors such as hypoglycemia could be important contributors to white matter abnormalities. Unfortunately, we were unable to assess these variables in our cohort due to the constraints of the revision period. We have now explicitly stated this limitation in the Limitations section with the following:
"Additionally, gestational diabetes, a known cause of foetal overgrowth, has been linked to a higher risk of respiratory distress syndrome, particularly in late preterm and term infants. Although our study did not assess the presence of gestational diabetes, it is possible that larger body size may contribute to an increased incidence of PWML through the development of relatively more severe respiratory morbidities."
This addition highlights the potential role of gestational diabetes and respiratory distress as contributing factors to the observed association, even though our study did not directly assess these variables.
- Line 40: Surfactant replacement has not been shown to affect neurological outcomes/ IVH.
- Response to the Reviewer: We appreciate this important clarification. While surfactant therapy improves RDS and may shorten the duration of mechanical ventilation, potentially reducing the risk of IVH, it does not have strong evidence comparable to antenatal steroids or delayed cord clamping. Based on your suggestion, we have removed this statement from the manuscript.
- It is surprising that they have detected this amount of injury in relatively late gestation infants.
- Response to the Reviewer: Thank you for your insightful comment. Indeed, it is intriguing that white matter injuries, including PWML, which have traditionally been reported mainly in infants born before 28 weeks of gestation, are also observed at similar rates in moderate and late preterm infants. To emphasize this novel finding, we have revised the Discussion section to strengthen the comparison with previous studies.
We sincerely appreciate the reviewer’s valuable feedback, which has helped improve the clarity and rigor of our manuscript.
Round 2
Reviewer 2 Report
Comments and Suggestions for Authors
Accepted